# Non-Alcoholic Fatty Liver Disease Defined by Fatty Liver Index and Incidence of Heart Failure in the Korean Population: A Nationwide Cohort Study

**DOI:** 10.3390/diagnostics12030663

**Published:** 2022-03-09

**Authors:** Byoungduck Han, Gyu Bae Lee, Sun Young Yim, Kyung-Hwan Cho, Koh Eun Shin, Jung-Hwan Kim, Yong-Gyu Park, Kyung-Do Han, Yang-Hyun Kim

**Affiliations:** 1Department of Family Medicine, Korea University College of Medicine, Seoul 02841, Korea; hbdfm@korea.ac.kr (B.H.); glee@korea.ac.kr (G.B.L.); chokh@korea.ac.kr (K.-H.C.); 2Department of Internal Medicine, Korea University College of Medicine, Seoul 02841, Korea; eug203@korea.ac.kr; 3International Healthcare Center, Korea University Anam Hospital, Seoul 02841, Korea; kirry0806@gmail.com; 4Department of Family Medicine, Eulji Hospital in Uijeongbu, Eulji University School of Medicine, Uijeongbu 11749, Korea; 12thrib@hanmail.net; 5Department of Medical Statistics, Catholic University College of Medicine, Seoul 06591, Korea; ygpark@catholic.ac.kr; 6Department of Statistics and Actuarial Science, Soongsil University, Seoul 06978, Korea

**Keywords:** heart failure, non-alcoholic fatty liver disease, fatty liver index, waist circumference, obesity

## Abstract

Fatty liver index (FLI) is a simple and useful index that evaluates non-alcoholic fatty liver disease (NAFLD), particularly in large epidemiologic studies. Heart failure (HF) is becoming a burden to public health as the global trend toward an aging society continues. Thus, we investigated the effect of FLI on the incidence of HF using large cohort data from the Korean National Health Insurance health database. **Methods and Results**: A total of 7,958,538 subjects aged over 19 years without baseline HF (men = 4,142,264 and women = 3,816,274) were included. Anthropometric and biochemical measurements were evaluated. FLI scores were calculated and FLI ≥ 60 was considered as having NAFLD. Hazard ratios (HRs) and 95% confidence intervals (CIs) for HF incidence were analysed using multivariable time-dependent Cox proportional hazard models. During a mean follow up of 8.26 years, 17,104 participants developed HF. The FLI components associated with the incidence of HF and FLI showed a causal relationship with HF; the FLI ≥ 60 group had a higher HR for HF (HR 1.493; 95% CIs 1.41–1.581) than the FLI < 30 group. Subgroup analysis showed that fatty liver (FLI ≥ 60) with age ≥ 65 years or women displayed higher HR for HF than fatty liver with age < 65 or men, respectively. An increase in FLI score significantly increased the HR for HF except for those with a FLI score change from <30 to 30–60. **Conclusion:** NAFLD defined by FLI and increase in FLI score were associated with the incidence of HF. Further detailed prospective studies are needed.

## 1. Introduction

The prevalence of heart failure (HF) is about 1% to 2% in developed countries and increases to more than 10% in individuals over 85 years of age [1]. The prevalence of HF continues to increase, and approximately 6.5 million people who were ≥20 years of age in the USA had HF between 2011 and 2014 compared with approximately 5.7 million individuals between 2009 and 2012 based on data from the National Health and Nutrition Survey (NHANES) [1]. In Korea, the prevalence of HF has doubled from 0.75% in 2002 to 1.53% in 2013 [2]. Once diagnosed with HF, patients suffer from both the medical and financial burden of illness. For instance, the hospitalization rates due to HF account for 2–3% of total healthcare costs [3]. Known risk factors for HF include hypertension (HTN), left ventricular hypertrophy, coronary heart disease, valvular heart disease, dyslipidaemia, albuminuria, smoking, low physical activity, glucose intolerance, and obesity [4,5,6].

Non-alcoholic fatty liver disease (NAFLD) is one of the common chronic liver diseases and has increased and affected a quarter of the whole global population [7]. In Asian countries, the pooled prevalence of NAFLD was about 27.4%, which was similar to Western countries [7,8]. In Korea, the prevalence of NAFLD was about 27% in the general population [9] and was 16.1% (21.6% in men, 11.2% in women) when diagnosed with ultrasnography [10]. NAFLD is a risk factor for major adverse cardiovascular events (MACE) and is a multisystem disease [11]. The gold standard diagnosis for NAFLD and non-alcoholic steatohepatitis (NASH) is biopsy, but being an invasive technique [12], various non-invasive indices have been developed especially for the purposes of public health check-up studies [13]. Fatty liver index (FLI) is one of the indices used to non-invasively evaluate fatty liver disease, which has been validated in Asian and Caucasian general populations [14,15,16]. Thus, many researchers are studying the Korean population using FLI [17,18].

Interestingly, a cohort study from Simon et al. demonstrated that biopsy-proven NAFLD in the Swedish population was correlated with a 65% increase in the risk of incident major adverse cardiac events, which includes HF, regardless of cardiometabolic risk factors (*n* = 10,422 and 13 years of follow-up) [19]. In addition, using a sample cohort from the national database (*n* = 308,578 and 5.4 years of median follow-up), Roh et al. demonstrated that FLI > 30 was related to higher risk of incident HF risk in healthy Koreans [20].

Evidence supporting the association between NAFLD and HF is available. However, the use of FLI in analysing the relationship between NAFLD and HF with a comprehensive population database in Korea has been limited in the past. We hypothesize that NAFLD, defined by FLI, is an independent risk factor for incident HF in the general population. Therefore, we examined the incidence of HF in subjects with or without NAFLD as defined by FLI using Korean National Health Insurance Service (NHIS) health check-up data.

## 2. Methods

### 2.1. The NHIS Database and NHIS Health Check-Up Program

The NHIS is an insurer that manages the National Health Insurance (NHI) program, which contains medical information that covers approximately 50 million Koreans [21]. Patient data that includes age, sex, area of residence, insurer payment coverage, and claims and deductions data sources can be obtained from the Health Insurance Review and Assessment (HIRA) service after permission. The HIRA database covers about 97.0% of the Korean population, and the details of the NHIS database have been previously described [22]. The NHIS provides a biannual or annual health check-up program that includes anthropometric measurements, blood tests, and health surveys including general health behaviours and past medical histories for all insured Koreans.

### 2.2. Subjects

We used the NHIS health check-up database from 2002 to 2017 and selected subjects aged ≥20 years old and those who had undergone the health check-ups in 2009. The year 2009 was selected because waist circumference measurements, a requisite for FLI calculation, was available starting from 2009. Additionally, we aimed to confirm that the participants were free of HF from 2002 and 2009, while maximizing the follow-up period. A total of 10,490,491 subjects were followed up to 31 December 2017. The following participants were excluded: subjects who drank heavily (drinking ≥30 g/day in men and ≥20 g/day in women) (*n* = 877,389), subjects who were diagnosed and claimed for HF prior to the index date (*n* = 210,433), subjects with a history of liver cirrhosis (*n* = 990,315), and subjects with missing data (*n* = 453,816). Finally, 7,958,538 subjects were included in this study (men = 4,142,264 and women = 3,816,274), and mean observation time was 8.26 ± 0.78 years. This study was approved by the Institutional Review Board of the Korea University Anam Hospital (IRB No. 2019AN0001), and permission for the use of health check-up data was granted by the NHIS (NHIS-2018-1-049).

### 2.3. General Health Behaviours

Self-reported questionnaires were obtained from the participants with respect to general health behaviours that included smoking status (no, ex-smoking, and current smoking), alcohol drinking (no and mild to moderate; drinking <30 g/day in men and <20 g/day in women), and regular exercise, defined as vigorous physical activity for at least 20 min/day. The frequency of exercise was categorized into 0, 1–4, ≥5 times/week.

### 2.4. Anthropometric and Biochemical Measurements

Anthropometric measurements such as weight (kg), height (cm), waist circumference (WC) (cm), systolic blood pressure (SBP) (mmHg), and diastolic blood pressure (DBP) (mmHg) were measured by trained examiners. Body mass index (BMI) (kg/m^2^) was calculated using weight (kg) divided by the square of the height (m). We defined obesity as a BMI ≥ 25 kg/m^2^, according to the World Health Organization recommendation for Asians [23].

The NHIS health check-up also includes laboratory tests, such as fasting blood glucose (mg/dL), total cholesterol (mg/dL), triglycerides (TG, mg/dL), high-density lipoprotein cholesterol (mg/dL), aspartate aminotransferase (AST, IU/L), alanine aminotransferase (ALT, IU/L), gamma-glutamyl transferase (GGT, IU/L), serum creatinine levels (mg/dL), and urine analyses. The Korean Association of Laboratory Quality Control warranted the quality of the laboratory tests, and the health check-up performing hospitals were certified by the NHIS.

### 2.5. Definition of Fatty Liver Index

We used the FLI, a surrogate marker of NAFLD, which was calculated based on a report by Bedogni et al., as follows: FLI = (e ^0.953 × loge (triglycerides) + 0.139 × BMI + 0.718 × loge (ggt) + 0.053 × waist circumference − 15.745^)/(1 + e ^0.953 × loge (triglycerides) + 0.139 × BMI + 0.718 × loge (ggt) + 0.053 × waist circumference − 15.745^) × 100 [14]. We categorized the study participants into 3 groups according to their FLI score (FLI < 30, 30 ≤ FLI < 60, and FLI ≥ 60) [24]. An FLI ≥ 60 was defined as having fatty liver [14].

### 2.6. Incidence of HF

Incidence of HF was defined according to the International Classification of Diseases (ICD-10) code I50 and an admission history with the HF code from 1 January 2009 to 31 December 2017 for each participant.

### 2.7. Definition of Chronic Diseases

Diabetes mellitus (DM) was defined as a fasting plasma glucose ≥126 mg/dL or at least one claim for antidiabetic medication prescription with the ICD-10 codes E11–E14. Hypertension was defined as an SBP ≥ 140 mmHg or a DBP ≥ 90 mmHg, or having at least one claim for antihypertensive medication prescription with ICD-10 codes I10–I15. Dyslipidaemia was defined by a total serum cholesterol level ≥ 240 mg/dL or having at least one claim for an anti-dyslipidaemic medication prescription with ICD-10 code E78. Cardiovascular disease (CVD) was identified when the subject gave an affirmative answer to the following question: “Do you have a history of acute myocardial infarction?”.

### 2.8. Statistical Analysis

Subject general characteristics are expressed as mean ± standard deviation (SD) for continuous variables and percentages (SD) for categorical variables. Variables which did not have a normal distribution (TG, AST, ALT, and GGT) are expressed as geometric means and interquartile ranges. The hazard ratios (HRs) and 95% confidence intervals (CIs) for the incidence of HF according to FLI group were assessed by multivariable Cox proportional hazard models using the FLI < 60 group as the reference group. Other variables were adjusted: age, sex, smoking, alcohol, and exercise in model 1 and age, sex, smoking, alcohol, exercise, HTN, dyslipidaemia, DM, and BMI in model 2. The HR and 95% CI for the incidence of HF according to each subgroup was also obtained by constructing a multivariable Cox model: age ≥ 65 years old, sex, history of cardiovascular diseases, and obesity. Interaction *p* for each subgroup was calculated for subgroup analysis. We also calculated the HR for the incidence of HF according to the change in FLI level during the four-year follow-up with stable FLI < 30 (no change in FLI < 30 during the four-year follow up) as the reference, using multivariable Cox proportional hazard models after adjusting for all covariables.

All statistical analyses were performed using SAS version 9.4 (SAS Institute, Cary, NC, USA), and a two-tailed *p*-value less than 0.05 was considered statistically significant.

## 3. Results

Table 1 shows the participant general characteristics according to the presence of fatty liver (FLI ≥ 60). Of the total 7,958,538 subjects, 17,104 had HF. Subjects with fatty liver were older and had increased BMI, WC, SBP, DBP, fasting glucose, total cholesterol, LDL-C, TG, AST, ALT, and GGT, but decreased HDL-C (all *p* < 0.001). The proportion of men, current smoking, mild alcohol drinking, DM, HTN, dyslipidaemia, and history of CVD was higher in the subjects with fatty liver, but the proportion of income (Q1) and regular exercise was lower in the subjects with fatty liver than subjects without fatty liver (all *p* < 0.001).

Figure 1 shows the incidence rate per 1000 persons and HR for the incidence of HF by FLI components. BMI was divided into five categories and WC was also divided into eight categories by 5 cm increase. TG and GGT were divided into two categories (TG ≥ 150 mg/dL or not and GGT ≥ 50 mg/dL or not). Incidence rate for HF increased as each FLI component increased, but BMI showed a J-shaped incidence rate for HF (Figure 1a). After adjusting for all covariates, as GGT and WC increased so did the HR for HF (Figure 1b). The highest HR of HF was associated with a WC of more than 110 cm in men and 105 cm in women (HR = 2.80 and 95% CI = 2.39–3.285). However, for BMI, subjects who were underweight (BMI < 18.5 kg/m^2^) had the highest HR for HF (HR = 1.718 and 95% CI = 1.678–1.758), and overweight (23 < BMI < 25 kg/m^2^) subjects had the lowest HR for HF (HR = 0.884 and 95% CI = 0.838–0.933) among the five BMI categories, and the HR for HF had a U-shaped curve (Figure 1b).

Table 2 shows the HR for the incidence of HF according to FLI level. We used FLI < 30 as a reference, and the HR of HF was 1.212 (95% CI = 1.077–1.167) for FLI 30–60 and 1.493 (95% CI = 1.41–1.581) for FLI ≥ 60 after adjusting for all covariables.

In subgroup analysis, women with fatty liver (FLI ≥ 60) showed a higher HR for HF than men (each HR = 1.481 and 1.276, respectively, and *p* for interaction = 0.001). The subjects who were older than 65 years also showed an increased HR compared with the subjects who were younger than 65 years (each HR = 1.353 and 1.26, respectively, and *p* for interaction <0.001). However, fatty liver without the history of CVD or without obesity had higher HRs for HF than those with a history of CVD or with obesity (*p* for interactions were 0.002 and 0.049, respectively) (Table 3).

In Figure 2, we calculated the HR for the incidence of HF according to change in FLI during the four-years of follow-up, with stable FLI < 30 (no change in FLI < 30 during the four-year follow-up) as the reference. After adjusting for all covariables, participants with a change in FLI showed significantly increased HR for HF, except for those with an FLI change from <30 to 30–60. An increase in FLI from <30 to ≥60 and 30–60 to ≥60 showed higher HRs (HR = 1.373 and 1.443, respectively) than those with a decrease in FLI (from 30–60 to <30 and from ≥60 to <30 and 30–60) (HRs = 1.161, 1.472, and 1.443, respectively). Even participants with a stable FLI 30–60 and stable FLI ≥ 60 showed increased HRs for HF and participants with a stable FLI ≥ 60 showed the highest HR for HF among all changes in FLI groups. (HR = 1.195 for stable 30–60 and 1.7 for stable ≥ 60).

## 4. Discussion

In this study, NAFLD as defined by FLI was associated with an increased HR for HF in the Korean population independent of conventional CVD risk factors, confirming the initial research hypothesis. The subgroup analysis revealed that subjects aged ≥65 years and the female sex had a higher increase in HR of HF than subjects aged <65 years and the male sex, respectively, by FLI group. Increase in FLI score correlated significantly with an increased HR for HF except for FLI score change from <30 to 30–60.

There has been a controversy over the relationship between NAFLD and CVD—namely, is NAFLD a precursor for future CVD or is it a passenger biomarker in the overall course of CVD development? The confusion arises partly because most NAFLD patients are also diabetic patients [25], and the relationship between NAFLD and LV structure/function or between NAFLD and subclinical diastolic function has been obscured in obese patients [26]. Such data may imply that NAFLD is on the passenger side in CVD development; nonetheless, researchers should not prematurely conclude that NAFLD and CVD are end organ damage due to metabolic syndrome [25].

A growing body of evidence is suggesting a strong association between NAFLD and cardiac complications, which may lead to HF development. Simon et al. have provided important evidence that biopsy-proven NALFD may act as a risk factor for new-onset HF [19]. Moreover, other studies that utilize noninvasive liver fibrosis scores in studying HF patients have reported higher prevalence of NAFLD in the patients with heart failure with preserved ejection fraction (HFpEF), further supporting the association between NAFLD and HF [27,28]. Patients with computed tomography-detected NAFLD reported a higher prevalence of coronary microvascular dysfunction and lower coronary flow reserve than those without NAFLD in a retrospective cohort study in patients without coronary artery disease and with preserved LV ejection fraction [29]. Patients with NAFLD displayed higher levels of impairment in heart rate variability, diastolic variability, and systolic variability than the controls in a study using magnetic resonance spectroscopy [30]. Arrhythmias can contribute to HF development, while permanent atrial fibrillation (AF), QTc prolongation, left anterior hemlock, and right bundle branch block have all been reported to be associated with NAFLD [31,32,33].

Cardiac remodelling is a well-understood and crucial component in the disease course of HF. Accumulating evidence suggests a link between NAFLD and new-onset HF with respect to cardiac remodelling. Many studies have reported an association between NAFLD and LV diastolic dysfunction or greater LV hypertrophy, regardless of metabolic risk factors [25,32,33]. NAFLD was related with subclinical cardiac remodelling independent of cardiometabolic risk factors [34].

Possible mechanisms underlying the increased risk of HF in NAFLD may involve hepatic and extrahepatic pathways [11]. The hepatic pathway could involve the progression of NAFLD and its intermediary factors contributing to increased risk of cardiovascular disease. Circulating atherogenic lipoprotein phenotype is one such component where pro-atherogenic lipid changes are observed in NAFLD [32]. Another component is the increased activation of the renin–angiotensin–aldosterone system (RAAS) in NAFLD [35]. RAAS activation is known to drive the progression of HF. In addition to that, hepatic mitochondrial dysfunction [36], and possibly myocardial mitochondrial dysfunction [37], coupled with the production of reactive oxygen species (ROS) may together contribute cardiac remodelling [38]. The extrahepatic pathway may deal with visceral adipose tissue and intestinal dysbiosis. Expanded visceral adipose tissue produces less serum adiponectin but secretes more pro-inflammatory cytokines, such as serum interleukin (IL)-6, which has been linked to subclinical atherosclerosis and atrial fibrillation [39,40]. Additionally, insulin resistance may promote the development and worsening of NAFLD and further encourage negative changes in coronary arteries to bring about cardiac disease [39]. Lipopolysaccharide, ethanol, short-chain fatty acids, and incretins from the gastrointestinal tract may lead to liver fibrosis [41,42,43,44] and directly affect the cardiovascular system. [45] Increase in Gram-negative bacteria can increase the endotoxin’s chance to enter the portal circulation and also cause inflammatory response in the liver [46]. Additionally, an increase in proteobacteria in the gut may promote higher production of endogenous alcohol and can perhaps contribute to liver fibrosis, inducing cardiac dysfunction [44].

Subgroup analyses with those with FLI ≥ 60 showed that both men and women exhibited increased risk of HF (1.28 and 1.48, respectively, *p* for interaction 0.001). Many epidemiological studies have confirmed that women are more likely to suffer from HFpEF than men. Researchers suspect that the female-specific pathway is involved; however, the precise mechanism is yet to be found [47,48]. Additionally, both obese (BMI ≥ 25) and non-obese subgroups with FLI ≥ 60 displayed a higher risk of HF (1.28 and 1.48, respectively, *p* for interaction 0.001). The obesity paradox may contribute to higher incidence of HF in the obese subgroup. In contrast, the “lean paradox” may explain the higher HR related to the non-obese subgroup. The lean paradox proposes that low BMI is associated with less physiologic reserve against cardiac mass loss, or cachexia, and therefore low body fat and low BMI are associated with more CVD outcomes [49,50].

This present study displays how NAFLD defined by FLI was associated with the increased risk of HF. Hence it is crucial that future studies focus on NAFLD treatment options, especially on how, and to what degree, NAFLD treatment can prevent HF and/or CVD. Current NAFLD treatment includes lifestyle modification, vitamin E, pioglitazone, and weight loss, including metabolic surgery [12]. However, new drugs to treat NAFLD are currently being developed to improve LV remodelling [51,52].

There are several limitations in our study. First, this study only included Korean participants, and results cannot be generalized to other ethnicities. Second, FLI is an indirect indicator of NAFLD, and NAFLD was not histologically confirmed in this study. Third, biomarkers that mediate FLI and HF—such as adiponectin, leptin, and dimethyl arginine—were not investigated [53,54]. Fourth, we did not distinguish NAFLD from NASH in subjects with fatty liver. Fifth, we defined HF based on the ICD-10 code and admission history for HF, so we could not determine the precise classification of HF (HFpEF or HFrEF) in this study. It is difficult to confirm the diagnosis of HF with echocardiography due to limitations in the NHIS health screening database. Nonetheless, the authors believe that the combination of the ICD-10 code and concurrent admission history is stringent and also the most viable means of confirmation, considering the study design. Similar definitions have been utilized in research regarding HF using the NHIS database [55,56]. Sixth, CVD events could have happened during the follow-up period prior to HF. Seventh, FLI was only applicable to the NHIS database starting from the year 2009 for the reason mentioned in Materials and Methods under the Subjects subheading, and it is possible some participants prior to enrolment may already have had NAFLD. It is worth noting that because there is a lack of treatment options for NAFLD physicians do not often enter the diagnostic code for NAFLD in South Korea. Lastly, NHIS data does not provide HbA1c values or a full list of medications for participants, so it is difficult to study any effects from medications or how well the comorbidities were controlled.

However, to the best of our knowledge, this is the first study to examine the relationship between NAFLD defined by FLI and incidence of HF in a comprehensive database of the Korean adult population. Second, we believe this is the first study to present how HF risk increases as NAFLD severity changes in the Korean population. Third, we conducted a nationwide study that involved a homogeneous participant group. Furthermore, we analysed various subgroups while adjusting for covariates that potentially affect HF incidence.

In conclusion, FLI was associated with increased HF incidence in national-scale epidemiological data. Although the FLI does not reflect the status of liver fibrosis, FLI could be used as one of the risk factors that predicts the risk of HF. Further detailed prospective studies that can confirm fatty liver by biopsy or at least with ultrasonography are needed to examine the relationship between fatty liver and HF. In addition, change in FLI was associated with increased HR for HF, thus developing medication or managing risk factors for NAFLD may be helpful in decreasing the incidence of HF in subjects with NAFLD.

## Figures and Tables

**Figure 1 diagnostics-12-00663-f001:**
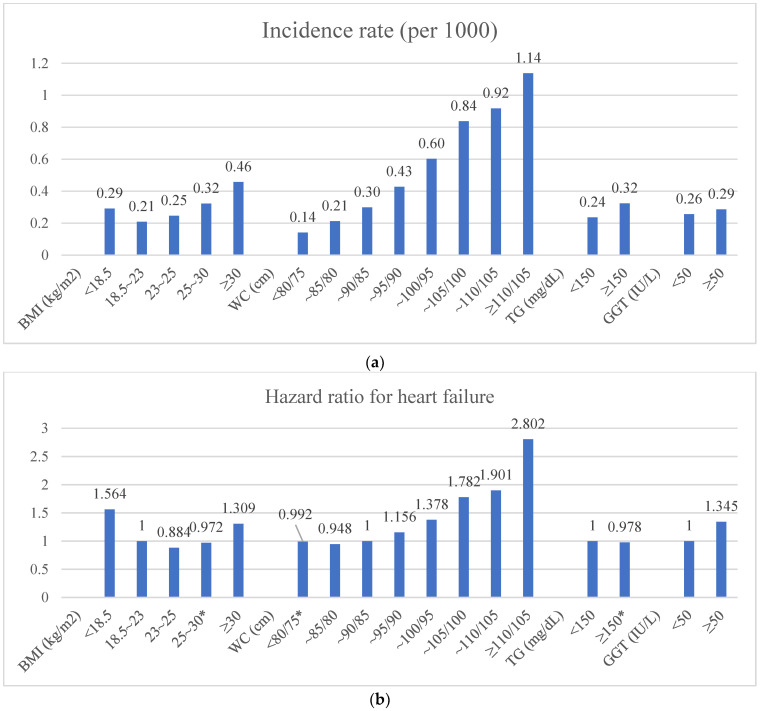
The incidence rate (per 1000 persons) and hazard ratio for heart failure by the components of the fatty liver index. * Statistically not significant. Adjusted for age, gender, smoking, drinking, exercise, diabetes mellitus, hypertension, dyslipidaemia, and BMI. BMI, body mass index; WC, waist circumference; TG, triglyceride; GGT, gamma-glutamyl transferase; Q, quintiles (Q1, Q2, Q3, Q4, and Q5). (**a**) Incidence rate for heart failure per 1000 persons, (**b**) Hazard ratio for the incidence of heart failure.

**Figure 2 diagnostics-12-00663-f002:**
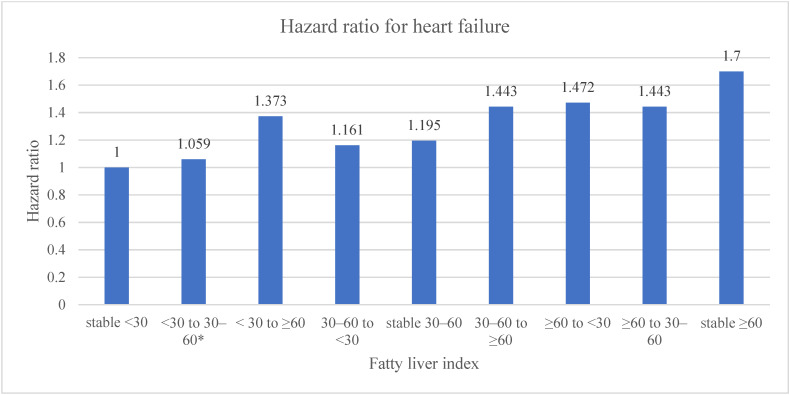
Change in fatty liver index and its hazard ratio for the incidence of heart failure during the four-year follow-up. * Statistically not significant. Adjusted for age, gender, smoking, drinking, exercise, diabetes mellitus, hypertension, dyslipidaemia, and body mass index.

**Table 1 diagnostics-12-00663-t001:** General characteristics of participants according to fatty liver index ≥ 60 or not.

Variables	FLI < 60	FLI ≥ 60	*p*
	(*n* = 7,065,123)	(*n* = 893,415)	
Age (years)	46.65 ± 14.2	47.23 ± 12.65	<0.001
Men (*n*, %)	3,410,822 (48.28)	731,442 (81.87)	<0.001
BMI (kg/m^2^)	23.04 ± 2.77	27.94 ± 2.98	<0.001
Waist circumference (cm)	78.13 ± 8.01	92.12 ± 6.74	<0.001
SBP (mmHg)	120.89 ± 14.63	129.77 ± 14.43	<0.001
DBP (mmHg)	75.27 ± 9.71	81.41 ± 9.85	<0.001
Fasting blood glucose (mg/dL)	95.09 ± 20.19	106.53 ± 31.23	<0.001
Total cholesterol (mg/dL)	192.93 ± 35.55	212.69 ± 38.74	<0.001
HDL-C (mg/dL)	56.22 ± 18.33	48.64 ± 19.81	<0.001
LDL-C (mg/dL)	114.07 ± 32.71	114.27 ± 38.65	<0.001
TG (mg/dL) *	100.06 (100.02–100.1)	229.04 (228.8–229.27)	<0.001
ALT (IU/L) *	19.12 (19.11–19.13)	37.3 (37.26–37.34)	<0.001
AST (IU/L) *	22.03 (22.02–22.03)	30.22 (30.19–30.24)	<0.001
GGT (IU/L) *	21.65 (21.64–21.66)	62.69 (62.6–62.78)	<0.001
Smoking (*n*, %)			<0.001
Non	4,637,491 (65.64)	345,997 (38.73)	
Ex	883464 (12.5)	179317 (20.07)	
Current	1,544,168 (21.86)	368,101 (41.2)	
Alcohol drinking (*n*, %)			<0.001
No	4,054,305 (57.38)	342,949 (38.39)	
Mild	3,010,818 (42.62)	550,466 (61.61)	
Regular exercise (*n*, %)	1,261,560 (17.86)	152,993 (17.12)	<0.001
Income (Q1) (*n*, %)	1,934,364 (27.38)	207,658 (23.24)	<0.001
Diabetes mellitus (*n*, %)	462,520 (6.55)	157,135 (17.59)	<0.001
Hypertension (*n*, %)	1,509,918 (21.37)	383,064 (42.88)	<0.001
Dyslipidaemia (*n*, %)	1,087,937 (15.4)	301,584 (33.76)	<0.001
History of cardiovascular disease (*n*, %)	105,968 (2.42)	17,017 (2.82)	<0.001
Heart failure (*n*, %)	14,414 (0.2)	2690 (0.3)	<0.001

Values presented as mean ± standard error or percentage (%) (standard error). * Geometric mean (95% confidence interval). BMI, body mass index; SBP, systolic blood pressure; DBP, diastolic blood pressure; HDL-C, high-density lipoprotein cholesterol; TG, triglyceride; ALT, alanine transaminase; AST, aspartate transaminase; and GGT, gamma-glutamyl transferase.

**Table 2 diagnostics-12-00663-t002:** Adjusted hazard ratios for the incidence of heart failure according to the level of fatty liver index.

Level of Fatty Liver Index					Model 1	Model 2
	N	Event	Duration	IR (per 1000)	HR (95% CI)	HR (95% CI)
<30	5,282,514	9265	43,688,896.61	0.212	1 (Ref.)	1 (Ref.)
30~60	1,782,609	5149	14,703,192.49	0.350	1.304 (1.26–1.349)	1.121 (1.077–1.167)
≥60	893,415	2690	7,351,950.24	0.366	1.939 (1.856–2.025)	1.493 (1.41–1.581)

**Model 1**: adjusted for age, gender, smoking, drinking, and exercise; **Model 2**: adjusted for age, gender, smoking, drinking, exercise, diabetes mellitus, hypertension, dyslipidaemia, and body mass index. IR, incidence rate; HR, hazard ratio; and CI, confidence interval.

**Table 3 diagnostics-12-00663-t003:** Adjusted hazard ratios for the incidence of heart failure in the subjects with fatty liver stratified to subgroups.

Subgroup	HR (95% Cl)	*p* for Interaction
Men	1.276 (1.193,1.364)	0.001
Women	1.481 (1.379,1.592)	
Age < 65	1.26 (1.159,1.371)	<0.001
Age ≥ 65	1.353 (1.273,1.438)	
No CVD	1.396 (1.316,1.481)	0.002
CVD	1.238 (1.083,1.416)	
No obesity	1.308 (1.169,1.463)	0.049
Obesity	1.203 (1.134,1.277)	

Adjusted for age, gender, smoking, drinking, exercise, diabetes mellitus, hypertension, dyslipidaemia, and body mass index. CVD, cardiovascular diseases; HR, hazard ratio; CI, confidence interval.

## Data Availability

Data request can be made to NHIS at the following website address. https://nhiss.nhis.or.kr/ accessed on 3 January 2022.

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
