# Peer review of "Non-Alcoholic Fatty Liver Disease Defined by Fatty Liver Index and Incidence of Heart Failure in the Korean Population: A Nationwide Cohort Study"

_diagnostics, 2022, doi:10.3390/diagnostics12030663_

Round 1

Reviewer 1 Report

  1. End the introduction by stating the hypothesis for the research, and include the specific research objectives to test your hypothesis that would logically follow the rationale for the research design. Justify the approach used in this study based on the hypothesis and specific objectives to test the hypothesis.
  2. Make certain that all footnotes for the figures and tables are correct and complete. Expand all used abbreviations.
  3. State if you accept or reject the research hypothesis in the discussion.
  4. There are some issues in References. Make certain that all of the references are correct (e.g. 10, 19 - full title of the journal; 9, 12 – abbreviation).

Author Response

Point 1

End the introduction by stating the hypothesis for the research, and include the specific research objectives to test your hypothesis that would logically follow the rationale for the research design. Justify the approach used in this study based on the hypothesis and specific objectives to test the hypothesis.

Response 1:

Thank you very much for your suggestion. We have added the research hypothesis and objectives in the introduction. We were not clear on the hypothesis before and we very much appreciate your advice. We hypothesize that NAFLD, defined by FLI, is an independent risk factor for incident HF in the general population. FLI is a non-invasive method to define NAFLD, which is particularly useful in population-based studies. Therefore, we examined the incidence of HF in subjects with or without NAFLD as defined by FLI using Korean National Health Insurance Service (NHIS) health check-up data.

Point 2:

Make­ certain that all footnotes for the figures and tables are correct and complete. Expand all used abbreviations.

Response 2:

Thank you very much for the advice. We have gone over all the footnotes and titles for the figures and tables and they are now complete and accurate. All abbreviations in the title of the figures/tables have been expanded as well.

Point 3:

State if you accept or reject the research hypothesis in the discussion.

Response 3:

We appreciate your suggestion. We have corrected the discussion to state that we accept our hypothesis

Point 4:

There are some issues in References. Make certain that all of the references are correct (e.g. 10, 19 - full title of the journal; 9, 12 – abbreviation).

Response 4:

We appreciate your kind suggestion. We have used MDPI endnote style to correct any inconsistencies in the references.

Reviewer 2 Report

Authors evaluated the association between FLI and HF using Korean nationwide cohort. Consequently, FLI was associated with HF independent of CVD history or obesity. This was an interesting report and well-written.

I think the article (ID1544487) is an interesting work and valuable in the novelty, originality and scientificity. But, in the article, the diagnosis of NAFLD (NASH) was not confirmed. Therefore the conclusion and the title should be corrected. Furthermore the etiology or mechanism of HF was unclear. I suggest this article can be accepted after minor revision in which the title and conclusion are corrected and discussion about the etiology or mechanism of HF is added in the discussion.

Author Response

Point 1

I think the article (ID1544487) is an interesting work and valuable in the novelty, originality and scientificity. But, in the article, the diagnosis of NAFLD (NASH) was not confirmed. Therefore the conclusion and the title should be corrected. Furthermore the etiology or mechanism of HF was unclear. I suggest this article can be accepted after minor revision in which the title and conclusion are corrected and discussion about the etiology or mechanism of HF is added in the discussion.

Response 1:

Thank you very much for your valuable advices. We have extensively gone over and edited the discussion and now the discussion includes in-depth discussion of the mechanism between NAFLD and HF. Additionally, since the diagnosis of NAFLD was not proven with biopsy, the conclusion and the tile have been corrected to clear any misunderstanding of the results. The new title is included in the manuscript and is as followed: NAFLD defined by FLI and incidence of HF in the Korean population: a nationwide cohort study

Reviewer 3 Report

  1. In the introduction, the authors was suggsted to review more previous manuscripts about and NAFLD and heart failure, as published in  J Am Coll Cardiol. 2022 Jan 18;79(2):180-191, ,ESC Heart Fail. 2021 Apr;8(2):789-798. Besides,the hypothesis about relationship between HF and HAFLD can be porposed.
  2. In the method, the auhtors used Korea NHI database between 2002 and 2017, but slected subjects   who underwent physiccal check up in 2009. Why??
  3. Any validation data of heart failure diagnosis  about the Korea NHI database.
  4. Any previous publications in Koreas using the Fatty liver index?
  5. At least how long between the index date and HF diagnosis  ?
  6. Any fluctuation of fatty liver index during observation period, and that may  change  the relationship between NAFLD and heart failure. Besides, the the patients before enrollment had the diagnosis of NAFLD?
  7. The first diagnosis of HF in the outpatient clinic  or during hospitalization? Any  other CV event before diagnosis of of HF?
  8. Any different relationship between NAFLD, diastolic heart failure and systolic heart failure
  9. Any data about the cardiac arrhythmia, lung disaease  (COPD), stroke, and these  comorbidites may also be related to HF
  10. Besides, renal function and inflammation, RBC, may also be important risk factors for the HF
  11. Any gender differecne ? In women , anu differecne before and after menopause? Any differecne of of young and middle age and olde age groups.
  12. How to expalin the  different odds ratio in the table 3, for example  women >men and non-obese >obese
  13. What about relationship between NAFLD and heart failure after adjusting sugar, blood pressure and cholesterol control rate  ?
  14. The effects of medications on the relationship between NAFLD and HF?
  15. Heavy drinkers exclued ? 

Author Response

Point 1

In the introduction, the authors was suggsted to review more previous manuscripts about and NAFLD and heart failure, as published in  J Am Coll Cardiol. 2022 Jan 18;79(2):180-191, ,ESC Heart Fail. 2021 Apr;8(2):789-798. Besides,the hypothesis about relationship between HF and HAFLD can be porposed

Response 1:

  • We deeply appreciate your kind advice and suggestion on the previous manuscripts related to NAFLD and heart failure. The manuscripts have helped us tremendously in formulating the clear hypothesis behind NAFLD and HF. We have extensively edited the introduction to include a clear hypothesis of the research paper.

Point 2:

In the method, the authors used Korea NHI database between 2002 and 2017, but slected subjects   who underwent physical check up in 2009. Why??

Response 2: Please provide your response for Point 2. (in red)

  • We appreciate your question. From the year 2002 through 2008, weight circumference (WC) measurements were not collected during the health screening check-up visits and WC is a required factor in calculating FLI. Therefore, in order to calculate FLI we had no choice but to use the health screening data starting from 2009. Moreover, we sought to maximize the follow-up period after screening visit therefore the health check-up participants from the year 2009 were selected. Additionally, we aimed to make sure the participants did not have HF between 2002 and 2009.

Point 3:

Any validation data of heart failure diagnosis  about the Korea NHI database.

Response 3:

  • Thank you again for the question. We defined heart failure as when two criteria were satisfied: a patient was billed with a main diagnosis with the ICD-10 code “I50” (Heart Failure) AND the patient was admitted at the same time. We believe it is a stringent criterion and one of the most practical means of defining HF patients in NHIS database. Past researches using NHIS database have used the similar definitions of HF. ( Lee, J.H. et al, Korean Circ J 2016, 46, 658-664 ; Seo, S.R. et al, Int J Environ Res Public Health 2020, 17)

Point 4:

Any previous publications in Koreas using the Fatty liver index?

Response 4:

  • Thank you for bringing up an important question. Many researchers are utilizing FLI in studying Korean population. Most importantly, Roh et al used a sample cohort from NHIS to study the association between higher FLI and HF (n= 0.3 million, 5 yr follow-up). However, our current study is superior because we utilize a comprehensive national database, rather than a sample cohort, and we follow-up on the 8 million participants over a longer period of time (8 yrs)
  • Here is a list of previous publications in Koreans using FLI : Roh, JH et al. BMC Cardiovasc Disord 20, 204 (2020). ; Cho EJ. et al. Diagnostics 2021, 11(12); Im, H.J. et al. Clin Res Hepatol Gastroenterol 2021, 45, 101526

Point 5:

At least how long between the index date and HF diagnosis?

Response 5:

  • After excluding the participants who were diagnosed and claimed for HF prior to index date (n=210,433), we have followed-up for an average duration of 8.26 years (0.027~9.0027years) and 17,104 participants developed HF during the time. I hope I am answering your question well here.

Point 6:

Any fluctuation of fatty liver index during observation period, and that may change the relationship between NAFLD and heart failure. Besides, the patients before enrollment had the diagnosis of NAFLD?

Response 6:

  • We appreciate your thoughtful advice. First of all, let us be clear that FLI is only applicable to health screening participants of NHIS starting from the year 2009.
  • In Figure 2, we show the hazard ratios for the incidence of HF according to change in FLI during the four-year of follow-up with stable FLI <30 (no change of FLI <30 during the four-year follow-up) as the reference. After adjusting for all covariables, all the participants with a change in FLI showed significantly increased HR for HF except for those with an FLI change from <30 to 30–60. An increase of FLI from < 30 to ≥60 and 30–60 to ≥60 showed increased HRs (HR=1.373 and 1.443, respectively) than those with a decrease in FLI (from 30–60 to <30 and from ≥60 to <30 and 30–60) (HRs= 1.161, 1.472, and 1.443, respectively). Even participants with a stable FLI 30–60 and stable FLI ≥60 showed increased HRs for HF and participants with a stable FLI≥ 60 showed the highest HR for HF among all changes in FLI groups. (HR = 1.195 for stable 30–60 and 1.7 for stable ≥60).
  • As for the diagnosis of NAFLD prior to the enrollment, yes, it is possible that some of the participants had already been diagnosed with NAFLD. We cannot find out whether or not they were given the diagnosis. It is a limitation of our research design/database. There is no treatment options for NAFLD and therefore physicians do not usually enter the diagnostic code for NAFLD in South Korea. The definitions using ICD-10 code have limitation in this aspect. Yet we believe more research is needed in this topic so physicians and patients can get the awareness of the clinical importance of NAFLD. We added this in the limitation section.

Point 7:

The first diagnosis of HF in the outpatient clinic or during hospitalization? Any other CV event before diagnosis of HF?

Response 7:

  • We appreciate your advice. We defined heart failure as when two criteria were satisfied: a patient was billed with a main diagnosis with the ICD-10 code “I50” (Heart Failure) AND the patient was admitted at the same time. Outpatient clinic visits were not considered in the definition of HF in the current study design.
  • CV events could have happened during the follow-up period before the diagnosis of HF. But we are unable to find out if there were any because we defined HF from the billing data in NHIS. It is a limitation to our study. We added this in the limitation section.

Point 8:

Any different relationship between NAFLD, diastolic heart failure and systolic heart failure

Response 8:

  • Thank you for the question. We defined HF with the ICD 10 codes from the billing data in NHIS and there is no classification of diastolic/systolic HF in the database. It is a limitation of the current study. We added this in the limitation section.

Point 9:

Any data about the cardiac arrhythmia, lung disaease  (COPD), stroke, and these  comorbidites may also be related to HF Besides, renal function and inflammation, RBC, may also be important risk factors for the HF

Response 9:

  • We appreciate your advice. We did re-analysis by multivariable Cox proportional hazard models with the cardiac arrhythmia, lung disaease (COPD), stroke, Hb, CKD. (Model 3.). And results of Model 3 shows tend to be similar to before adjusted for COPD, Stroke, Cardiac arrhythmia, Hb and CKD.
    • cardiac arrhythmia, lung disaease (COPD), and stroke defined according to ICD-10 code I49, J41-44, and I63, respectively. CKD is defined as a eGFR<60.

Level of fatty liver index

Model 1

Model 2

Model 3

N

Event

Duration

IR (per 1000)

HR (95% CI)

HR (95% CI)

HR (95% CI)

<30

5282514

9265

43688896.61

0.212

1 (Ref.)

1 (Ref.)

1 (Ref.)

30~60

1782609

5149

14703192.49

0.35

1.304 (1.26-1.349)

1.121 (1.077-1.167)

1.133 (1.088-1.179)

≥60

893415

2690

7351950.24

0.366

1.939 (1.856-2.025)

1.493 (1.41-1.581)

1.531 (1.446-1.621)

Model 3: adjusted for age, gender, smoking, drinking, exercise, diabetes mellitus, hypertension, dyslipidaemia, and body mass index, COPD, Stroke, Cardiac arrhythmia, Hb and CKD

Point 10:

Any gender differecne? In women , any differecne before and after menopause? Any differecne of young and middle age and older age groups.

Response 10:

  • Thank you for the question. We did subgroup analysis for menopause as follow. But we cannot define menopause with our data, therefore, the analysis was divided based on the age of 50 which is known as average age of menopause in Korean women. Women who were younger than 50 years-old also showed an increased HR compared to women who were older than 50 years-old (each HR=2.517 and 1.437, respectively and p for interaction <0.001). In this study, we did not analyze the relationship between FLI and the incidence of heart failure according to sex and age in detail, but we plan to analyze this in more detail in future studies.

Subgroup

HR (95% Cl)

p for interaction

< 50 yrs

2.517 (1.86, 3.407)

<0.001

≥50 yrs

1.437 (1.337, 1.545)

Point 11:

How to expalin the  different odds ratio in the table 3, for example  women >men and non-obese >obese

Response 11:

  • We appreciate your input. Both men and women exhibited increased risk of HF (1.28 and 1.48, respectively, p for interaction 0.001) Many epidemiological studies have confirmed that women are more likely to suffer from HFpEF than men. However, the precise mechanism between HFpEF and women is yet to be found out. Authors suspect that female-specific pathway may involve NAFLD but no exact mechanism has been proposed.
  • Both obese and non-obese subgroups with increased FLI showed higher risk of HF (HR 1.2 and 1.31, respectively, p for interaction 0.049). Obesity paradox may contribute to higher incidence of HF in the obese subgroup. In contrast, “lean paradox” may explain the higher HR related to the non-obese subgroup. Lean paradox proposes that low BMI is associated with less physiologic reserve against cardiac mass loss, or cachexia, and therefore low body fat and low BMI are associated with more CVD outcomes. The above replies have been added to the manuscript.

Point 12:

What about relationship between NAFLD and heart failure after adjusting sugar, blood pressure and cholesterol control rate

Response 12:

  • Thanks for the question. When we calculated the HR for HF incidence in Table 2, we have adjusted for any confounding effects from having the prior diagnoses of diabetes mellitus, hypertension, and dyslipidaemia. Only the history of being diagnosed with DM, HTN, Dyslipidaemia were considered for during the analyses. We did not adjust for how well the underlying comorbidities were controlled. The database does not collect HbA1c level so we will further look into this question on our next research. It is a limitation in our study and we now have added this limitation section in the manuscript.

Point 13:

The effects of medications on the relationship between NAFLD and HF?

Response 13:

  • I am perhaps making an assumption here. I hope I am answering your question correctly. We cannot look into the full list of prescription medications of the participants to find out whether medications that are likely to contribute to HF were prescribed (e.g. Thiazolidinediones for diabetics). NHIS data does not have such information. It is a limitation in our study and we edited the manuscript accordingly.

Point 14:

Heavy drinkers exclued?

Response 14:

  • Thank you for the question. Self-reported questionnaire on alcohol drinking was used to classify participants and we only included subjects who are “no-drinking” or “mild to moderate” (drinking <30 g/day in men and <20g/day in women). Liver cirrhosis patients were excluded prior to enrollment.

Round 2

Reviewer 1 Report

The article can be published in the current version.

Author Response

Thank you for your comment. 

Reviewer 3 Report

Generally, I had no more comments, and the authors had made great effort to response  to my  questions. However, although the authors had made  response to the  questions,  it is suggested that the authors  can in the cover letters to make a clear statement  where he had made revisions, such as in the introduction  or result section or discussion  section, and at what paragraph?? , line??, page??
